# Does LSD confer lasting psychological resilience? an investigation of naturalistic users experiencing job loss

Benjamin A. Korman [1,2,3]*

1 Chair of Organizational Behavior, University of Konstanz, Konstanz, Germany, 2 Cluster of Excellence "The Politics of Inequality", University of Konstanz, Konstanz, Germany, 3 Department of Competencies, Personality, Learning Environments, Leibniz Institute for Educational Trajectories, Bamberg, Germany

* benjamin.korman@uni-konstanz.de, benjamin.korman@lifbi

## Abstract

Recent studies on classic psychedelics have suggested that their use is associated with psychological strengths and resilience, thereby conferring users a type of psychological protection relative to non-users. However, this idea has been brought into question by recent findings suggesting that lifetime users of lysergic acid diethylamide (LSD) report worse mental health during stressful experiences. The current study addresses these mixed findings by examining whether LSD use prior to a stressful experience buffers against the psychological distress experienced in the wake of the stressful experience. This study draws on openly-available data from the National Survey on Drug Use and Health (2008–2019) on 5,067,553 (weighted) unemployed, job seeking individuals experiencing job loss. Using purposeful respondent exclusion criteria to establish temporal precedence of the variables under investigation, this study offers a straightforward test of whether LSD use confers psychological resilience to naturalistic users. LSD use prior to job loss was associated with a higher likelihood of severe psychological distress following job loss, regardless of whether sociodemographic variables were controlled for or not. In sum, this study fails to find evidence for LSD-conferred psychological resilience in naturalistic users in the wake of a stressful experience.

## Introduction

Classic psychedelics are serotonin receptor agonists with psychoactive properties [1]. They include lysergic acid diethylamide (LSD), psilocybin (the psychoactive compound in magic mushrooms), mescaline (the psychoactive compound in the peyote and San Pedro cacti), and N,N-Dimethyltryptamine (DMT). Their effects on mental health have been the topic of an exponentially growing number of studies within the past two decades [2], and in recent years it has been suggested that the naturalistic use of classic psychedelics could be linked to psychological resilience [3–6], an attribute related to maintaining/regaining positive mental health during/following stressful experiences [7,8]. Initial support for this idea has come in various forms, including survey studies comparing the personality traits of naturalistic classic

**Data Availability Statement:** The data underlying the results presented in the study are available from the National Survey on Drug Use and Health (https://nsduhweb.rti.org/). Furthermore, the Stata code used to run the analyses is available on the

Open Science Framework (https://osf.io/dhbpt/?
view_only=e7a56f45aaf47cea91894955b0e89fa).

**Funding:** This work was supported by the
Deutsche Forschungsgemeinschaft (DFG-German
Research Foundation) under Germany's Excellence
Strategy (Grant Number EXC2035/1-390681379).
The funders had no role in study design, data
collection and analysis, decision to publish, or
preparation of the manuscript.

**Competing interests:** The author has declared that
no competing interests exist.

psychedelic users with non-users [3,4], as well as associations between naturalistic psychedelic use and various mental health outcomes in the United States [9–13] and other countries [4,6].

In their study comparing the psychological traits of users with non-users, Brasher and colleagues [3] found that, relative to non-users, psychedelic users reported higher scores on a variety of favorable psychological traits including equanimity, self-kindness, and self-transcendence, among others. The authors referred to these traits more generally as "psychological strengths", asserting that psychedelic users had an adaptive psychological profile while implying their unique psychological resilience relative to non-users. Similarly, upon finding that lifetime psychedelic use correlated with the personality traits "stability" and "plasticity" in their Argentinian sample, Cavanna and colleagues [4] interpreted these results as suggesting an "enhanced resilience and well-being in the light of challenging situations" (p. 91). Favorable associations between naturalistic classic psychedelic use and various measures of mental health and mental health related behaviors (e.g., psychological distress; suicidal planning, attempting suicide) have also begun to be framed as "protective" [6,9–13], implying that naturalistic use of classic psychedelics may make users more resilient to future stressors.

Each of these studies, however, has methodological weaknesses making the proposed link between LSD use and resilience tenuous. To start, research linking psychedelic use to psychological traits associated with resilience [3,4] cannot reveal whether differences between users and non-users are a result of actual use, or simply accompany an interest in, or desire to use, psychedelics. Second, the population studies suggesting that naturalistic classic psychedelic use may be protective against poor mental health [6,9–13] did not take into account whether this use occurred prior to, or following, stressful experiences. As such, they cannot speak to whether naturalistic classic psychedelic use is protective against the negative mental health outcomes of stressful experiences (i.e., confers resilience), or rather helps users recover following them. Taken together, although classic psychedelics have begun to be introduced as a potential source of psychological resilience, solid empirical evidence for this claim is lacking.

Recent findings from two survey studies have also brought this idea of psychedelically-induced psychological resilience in naturalistic users into question. First, Korman [14] reported that lifetime classic psychedelic use (specifically, that of LSD and psilocybin) is associated with slightly greater, not fewer, symptoms of psychological distress in unemployed job seekers located in the United States. Second, Bălăeț and colleagues [15] reported worse moods and lower resilience among psychedelic users relative to non-users during the COVID-19 pandemic in a sample from the United Kingdom. Although these two studies were not designed to specifically test whether classic psychedelic use prior to stressful experiences (e.g., job loss, enforced lockdown) offers psychological resilience in the wake of such experiences, they nonetheless introduced the novel perspective that prior classic psychedelic use can be associated with negative mental health outcomes during stressful experiences.

It is important to note that these two studies also suffer from methodological weaknesses making it difficult to determine whether prior naturalistic classic psychedelic use helps or harms users' mental health in the wake of stressful experiences. First, and similar to the aforementioned population studies linking lifetime classic psychedelic use to mental health outcomes, Korman's study [14] on unemployed job seekers did not consider when respondents last used classic psychedelics, and thus the reported findings may be explained by naturalistic classic psychedelic use while unemployed (i.e., in the midst of a stressful experience and, hence, with an unsuitable mindset) [16]. Second, the study by Bălăe and colleagues [15] focused on psychedelic use during the COVID-19 pandemic. As such, they did not report whether use of classic psychedelics prior to the pandemic conferred resilience to users following the start of the pandemic.

Taking the weaknesses of this previous work into consideration, the current study aims to examine whether classic psychedelic use confers users with psychological resilience. Specifically, it examines whether naturalistic use of LSD prior to job loss (a stressful experience filled with uncertainty) [17,18] is associated with a lower likelihood of severe psychological distress following job loss. In doing so, it works towards a clearer understanding of when and how the naturalistic use of classic psychedelics is related to mental health outcomes in the working population.

## Materials and methods

### Study data and sample

The current study uses publicly available data from the National Survey on Drug Use and Health on the United States (U.S.) civilian, non-institutionalized population [19]. Survey respondents were randomly selected from across the U.S. and compensated $30 for participating. Informed consent was obtained from all respondents and the NSDUH data collection was approved by the institutional review board at RTI International. Additional information regarding the sampling and data collection methods, along with the data itself, can be found on the NSDUH website (https://nsduhweb.rti.org/). Secondary analysis of the NSDUH data was exempt from review by the ethics committee of the University of Konstanz.

The present research sample was limited to respondents aged 18 or older who had participated in the survey in the years from 2008 to 2019. The restriction of data to the survey years between 2008 and 2019 was dictated by the fact that certain covariates were only available for these survey years.

Importantly, the study sample was also limited to respondents currently experiencing job loss, operationalized as respondents having been employed within the past year but, at the time of the survey, were unemployed and searching for work. Based on the design of the NSDUH survey, respondents who indicated being unemployed and searching for work simultaneously ruled out that their unemployment was planned (e.g., retirement) or due to an accident (e.g., disability). Although information was unavailable as to the specific reason for respondents' transition from being a job holder to an unemployed job seeker (e.g., whether they were fired, quit, or had a work contract that ended), regardless of the reason, this transition represents a stressful experience linked to various negative mental health outcomes [20].

**Temporal precedence.** To ensure temporal precedence of the study variables, that is that LSD users' last use was prior to their job loss (see Fig 1), purposeful exclusion criteria were applied to the NSDUH dataset. First, the current study's sample excluded LSD users who had used LSD within the past year. Second, due to the fact that the NSDUH does not ask respondents when their last use of other classic psychedelics (i.e., psilocybin, mescaline, and DMT) occurred, respondents reporting lifetime use of any of these other classic psychedelics were also excluded. This was to ensure that respondents' likelihood of having severe psychological distress could not be accounted for by any classic psychedelic use following their job loss.

| Event | LSD use | Job loss | Severe psychological distress |
|---|---|---|---|
| *Time* | *Prior to 12 months ago* | *Within past 12 months* | *Within past 30 days* |

**Fig 1. Temporal outline of study variables.**

Survey data were available for 15,854 respondents (weighted = 5,067,553), of whom 48% (weighted) were White, 53% (weighted) were male, and 44% (weighted) had at least some college experience.

## Measures

**Independent variable.** The independent variable was prior LSD use (0 = no, 1 = yes) and 520 respondents (4.2% weighted) reported having used LSD (but not within the past year).

**Dependent variable.** The dependent variable was psychological distress experienced in the past 30 days as measured by the 6-item Kessler Psychological Distress Scale [21]. Respondents indicated to what extent they felt the following in the past 30 days: 1) nervous, 2) hopeless, 3) restless or fidgety, 4) sad or depressed that nothing could cheer you up, 5) that everything was an effort, and 6) down on yourself, no good, or worthless. Respondents indicated their score for each item on a scale from 0 ("none of the time") to 4 ("all of the time") and these individual item scores were summed to calculate a total scale score. Individuals with a total score of 13 or higher were coded as having experienced severe psychological distress, whereas those whose score was below 13 were coded as not having experienced severe psychological distress (0 = did not experience severe psychological within the past 30 days, 1 = experienced severe psychological within the past 30 days). In the study sample, 1,872 respondents (10% weighted) reported having severe psychological distress within the past 30 days.

**Covariates.** The following were included as covariates: Age in years (18–25, 26–34, 35–49, 50–64, 65 or older), sex (male or female), ethnoracial identity (non-Hispanic White, Hispanic African American, non-Hispanic Native American/Alaska Native, non-Hispanic Native Hawaiian/Pacific Islander, non-Hispanic Asian, non-Hispanic more than one race, or Hispanic), educational attainment (less than high school, high school graduate, some college or Associate's degree, or college graduate), marital status (married, widowed, divorced/separated, or never been married), annual household income (less than $20,000, $20,000–49,999, $50,000–74,999, $75,000 or more), self-reported engagement in risky behaviour (never, seldom, sometimes, or always), overall health (poor, fair, good, very good, or excellent), and health insurance (no or yes). Each covariate was coded separately.

## Statistical analyses

Stata/SE 16.1 was used to run all analyses [22] and statistical significance was set at $p < .05$ (two-tailed). Sampling weights generated by the NSDUH to account for respondents' probability of selection and adjusted for their non-response were incorporated into the analyses. This was done using the Stata survey "svy" command and it ensured consistency between the sampled population and population estimates from the U.S. Census Bureau. As recommended by the Center for Behavioral Health Statistics and Quality [23], sample weights were calculated by dividing the original person-level sample weights by the number of combined datasets (i.e., survey years) used in the current study.

Given the binary nature of the dependent variable severe psychological distress, logistic regression was implemented. Variance inflation factors were calculated for all predictor variables, with each below 2.5. This suggested that multi-collinearity did not pose a problem for the subsequent statistical analyses.

## Results

### Main analyses

Before accounting for sociodemographic and other covariates, prior LSD use was associated with a significantly higher likelihood of having experienced severe psychological distress in the past month (see Table 1: Model 1). This association was robust to the inclusion of sociodemographic and other covariates in the statistical model (see Table 1: Model 2). Specifically, the results indicate that individuals who had used LSD prior to their job loss were roughly 1.6 to 1.7 times more likely to report subsequent severe psychological distress compared to those who had not used LSD prior to their job loss.

### Supplemental analyses

Supplemental analyses were conducted to determine whether prior LSD use buffered against non-severe psychological distress (i.e., psychological distress operationalized as a total score below 13 on the 6-item Kessler Psychological Distress Scale). These supplemental analyses used a subsample of that used in the main analyses, that is, only individuals without severe psychological distress (N = 13,982). Non-severe psychological distress ($M = 3.91$, $SD = 3.59$, weighted) was a non-normally distributed continuous variable and, for this reason, negative binomial regression was implemented. Similar to the main analyses, prior LSD was associated with greater symptoms of (non-severe) psychological distress within the past month (see Table 2: Models 1 and 2), regardless of whether sociodemographic and other covariates were accounted for in the statistical model.

**Table 1. Predicting severe psychological distress with prior LSD use.**

| Predictors | Model 1 | | | Model 2 | | |
|---|---|---|---|---|---|---|
| | aOR | SE | p | aOR | SE | p |
| Prior LSD use (0 = no, 1 = yes) | 1.727 | 0.314 | .003 | 1.558 | 0.297 | .022 |
| Age | | | | 0.800 | 0.043 | < .001 |
| Sex (1 = male, 2 = female) | | | | 1.752 | 0.163 | < .001 |
| Ethnoracial identity | | | | | | |
| African American | | | | 0.775 | 0.086 | .023 |
| Native American/Alaska Native | | | | 0.850 | 0.223 | .537 |
| Native Hawaiian/Pacific Islander | | | | 2.353 | 2.273 | .378 |
| Asian | | | | 0.723 | 0.194 | .230 |
| More than one race | | | | 0.678 | 0.150 | .082 |
| Hispanic | | | | 0.622 | 0.074 | < .001 |
| Educational attainment | | | | 0.966 | 0.042 | .425 |
| Marital status | | | | | | |
| Widowed | | | | 1.691 | 0.650 | .174 |
| Divorced or separated | | | | 1.231 | 0.199 | .200 |
| Never been married | | | | 1.341 | 0.176 | .028 |
| Annual household income | | | | 0.933 | 0.039 | .103 |
| Risky behavior | | | | 1.560 | 0.075 | < .001 |
| Overall health | | | | 0.540 | 0.024 | < .001 |
| Health insurance (0 = no, 1 = yes) | | | | 1.083 | 0.084 | .306 |
| Constant | 0.106 | 0.004 | < .001 | 0.525 | 0.158 | .034 |

*Notes*: N = 15,854 (weighted = 5,067,553); *aOR*: Adjusted odds ratios; *SE*: Standard error; the comparison group for ethnoracial identity is White; the comparison group for marital status is married.

**Table 2.  Predicting non-severe psychological distress with prior LSD use.**

| Predictors | Model 1 | | | Model 2 | | |
|---|---|---|---|---|---|---|
| | B | SE | p | B | SE | p |
| Prior LSD use (0 = no, 1 = yes) | 0.175 | 0.062 | .006 | 0.178 | 0.068 | .010 |
| Age | | | | -0.121 | 0.022 | < .001 |
| Sex (1 = male, 2 = female) | | | | 0.187 | 0.027 | < .001 |
| Ethnoracial identity | | | | | | |
| African American | | | | -0.009 | 0.036 | .808 |
| Native American/Alaska Native | | | | -0.217 | 0.135 | .111 |
| Native Hawaiian/Pacific Islander | | | | 0.061 | 0.181 | .735 |
| Asian | | | | 0.039 | 0.062 | .535 |
| More than one race | | | | -0.098 | 0.099 | .325 |
| Hispanic | | | | -0.196 | 0.041 | < .001 |
| Educational attainment | | | | 0.040 | 0.018 | .024 |
| Marital status | | | | | | |
| Widowed | | | | 0.020 | 0.148 | .894 |
| Divorced or separated | | | | 0.165 | 0.049 | .001 |
| Never been married | | | | 0.105 | 0.043 | .016 |
| Annual household income | | | | -0.017 | 0.012 | .151 |
| Risky behavior | | | | 0.192 | 0.017 | < .001 |
| Overall health | | | | -0.136 | 0.015 | < .001 |
| Health insurance (0 = no, 1 = yes) | | | | 0.001 | 0.029 | .969 |
| Constant | 1.355 | 0.014 | < .001 | 1.611 | 0.111 | < .001 |

*Notes*. $N$ = 13,982 (weighted = 4,567,562); *B*: Beta coefficient; *SE*: Standard error; the comparison group for ethnoracial identity is White; the comparison group for marital status is married.

## Discussion

Despite the ongoing revival in research on classic psychedelics [24], few studies have investigated how prior naturalistic classic psychedelic use is associated with mental health outcomes during later stressful experiences. This is a valuable area of research given the plethora of correlational studies associating lifetime classic psychedelic use with better mental health [9,12,25]. The current study takes an important step forward in determining whether these previously reported, optimistic findings derive from psychological resilience conferred by LSD use, an idea that has recently gained ground in the literature. Alas, the current study failed to find evidence that LSD grants naturalistic users psychological resilience to future stressors. Instead, LSD users were found to have a higher likelihood of reporting severe psychological distress following job loss compared to non-users. Comparable results from a supplemental analysis were obtained for respondents reporting non-severe levels of psychological distress.

The current study's findings are consequential considering the decades-long rise in LSD use among the general U.S. population [26,27]. Although this rise may be due to decreasing perceptions of risks associated with LSD use [27], it may also, in part, be due to LSD's use as a means of personal wellness [28], an idea promoted by overstated claims made by the rapidly growing psychedelic industry [2,29]. However, considering the current study's findings, LSD use should be advised against as a means for healthy individuals to gain psychological resilience. In addition, researchers should be wary of implying that LSD use is associated with psychological strengths or resilience as stronger support for this claim is needed.

The current study's findings contrast with previous work linking classic psychedelic use to an array of positive psychological traits associated with resilience [3]. This discrepancy may stem from previous studies' reliance on comparisons between individuals with previous LSD experience and those without. Such studies provide limited insight on the potential resilience-conferring effects of classic psychedelics because interest in classic psychedelics and a desire to use them (or some correlated third variable) may explain these relationships, as opposed to the actual use and long-term impact of these substances. Future studies could instead compare the psychological profiles of classic psychedelic users with those interested in, and intending to use, but who have not yet had a classic psychedelic experience. Such an approach could offer a clearer understanding as to why lifetime classic psychedelic users score higher on psychological traits associated with resilience, yet do not report less psychological stress following a stressful event (e.g., job loss; enforced lockdown) compared to non-users.

### Limitations

Several important limitations of the current study should be noted. First, it did not take into account the temporal distance between respondents' last use of LSD and their job loss. Thus, the current study cannot speak to whether LSD confers short-term, as opposed to long-term, resilience to users and future studies will be needed to determine this. Second, the current study is limited by its focus on job loss as a type of stressful event. Therefore, it cannot definitively answer the question of whether LSD confers resilience during other types of stressful events. Given that initial findings suggest classic psychedelics increase social connectedness [30,31], future studies could test for LSD-conferred resilience in individuals experiencing stressful events related to their social life, such as the ending, or loss, of a close relationship (e.g., divorce or death of a loved one). Lastly, the current study was limited to the study of psychological resilience conferred by LSD specifically. Thus, it cannot rule out the potential resilience-conferring effects of other classic psychedelics (i.e., psilocybin, mescaline, or DMT), and future studies will be needed to determine whether they, in contrast to LSD, offer users protection from the negative psychological impacts of stressful experiences.

As a final note, although the current study finds no evidence for psychological resilience stemming from naturalistic use in the working population, its findings are not meant to contest classic psychedelics' potential therapeutic use in clinical populations. Initial work has theorized that psychedelic-assisted therapy may increase resilience in clinical populations [32,33] and this line of research is becoming increasingly important given the growing number of individuals reporting mental health problems (e.g., major depressive episode) [34]. However, healthy individuals hoping to shield themselves from future psychological stressors should not look to LSD to do so.

### Conclusions

Although classic psychedelics have recently been discussed in the literature in terms of the psychological strengths, resilience, and protective associations related to their use, little is known regarding whether prior use is associated with better mental health outcomes in naturalistic users dealing with stressful experiences. This study addresses this gap in understanding by demonstrating that LSD does not confer long-term, psychological resilience in naturalistic users experiencing job loss.

### Author Contributions

**Conceptualization:** Benjamin A. Korman.

**Data curation:** Benjamin A. Korman.

**Formal analysis:** Benjamin A. Korman.

**Investigation:** Benjamin A. Korman.

**Visualization:** Benjamin A. Korman.

**Writing – original draft:** Benjamin A. Korman.

**Writing – review & editing:** Benjamin A. Korman.

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
