## [Decision Letter · Decision Letter 0]

23 Feb 2024

PONE-D-23-22708

Does LSD confer lasting psychological resilience? An investigation of naturalistic users experiencing job loss

PLOS ONE

Dear Dr.  Korman,

Thank you for submitting your manuscript to PLOS ONE. After careful consideration, we feel that it has merit but does not fully meet PLOS ONE’s publication criteria as it currently stands. Therefore, we invite you to submit a revised version of the manuscript that addresses the points raised during the review process.

We look forward to receiving your revised manuscript.

Kind regards,

Anthony A. Olashore, MBCHB, PhD, FWACP

Academic Editor

PLOS ONE

Journal Requirements:

Reviewers' comments:

Reviewer's Responses to Questions

**Comments to the Author**

1. Is the manuscript technically sound, and do the data support the conclusions?

Reviewer #1: Yes

Reviewer #2: Yes

2. Has the statistical analysis been performed appropriately and rigorously? 

Reviewer #1: I Don't Know

Reviewer #2: Yes

3. Have the authors made all data underlying the findings in their manuscript fully available?

Reviewer #1: Yes

Reviewer #2: Yes

4. Is the manuscript presented in an intelligible fashion and written in standard English?

Reviewer #1: Yes

Reviewer #2: Yes

5. Review Comments to the Author

Reviewer #1: This is an interesting and important study. In short, the article is well constructed and well written.

I have a minor suggestion. The methodology section describes "Survey respondents were randomly selected from across the U.S. and compensated $30 for participating."

According to me, this make slight confusion with the statement "Using purposeful respondent exclusion criteria, about purposeful exclusion criteria". Is this necessary to highlight the temporal precedence? Cause every study has some exclusion criteria with the logic. Can the statement be modified in any other format?

Did the compensation of $30 for participating hamper the authenticity of the study? I think, a statement regarding this issue would be better.

Reviewer #2: The manuscript is technically sound with conclusions clearly demonstrable from the data presented, the statistical analysis is rigorous and appropriate, the data is from a publicly available dataset, and is written in fully intelligible standard English.

6. PLOS authors have the option to publish the peer review history of their article (what does this mean?). If published, this will include your full peer review and any attached files.

Reviewer #1: No

Reviewer #2: **Yes: **Olorunfemi Oladotun Ogunwobi

---

## [Author Response · Author response to Decision Letter 0]

6 Mar 2024

Review Comments to the Author

Reviewer #1: This is an interesting and important study. In short, the article is well constructed and well written.

//Author response: Thank you for your kind words and I’m happy to hear that you found the study interesting and important.//

I have a minor suggestion. The methodology section describes "Survey respondents were randomly selected from across the U.S. and compensated $30 for participating."

According to me, this make slight confusion with the statement "Using purposeful respondent exclusion criteria, about purposeful exclusion criteria". Is this necessary to highlight the temporal precedence? Cause every study has some exclusion criteria with the logic. Can the statement be modified in any other format?

Did the compensation of $30 for participating hamper the authenticity of the study? I think, a statement regarding this issue would be better.

// Author Response: Thank you for bringing this lack of clarity regarding the exclusion criteria applied in the analyses to my attention. I can see why my sole reference to the “purposeful exclusion criteria” in the original submissions’ abstract was unclear and needed modifying. To be clear, because the study data draw on a cross-sectional survey panel, purposeful exclusion criteria are indeed necessary to establish temporal precedence among the variables in my study (i.e., LSD use and psychological distress as shown graphically in Fig 1). 

It is true that every study has some type of exclusion criteria. However, what I now try to highlight in my revised Abstract and Method section (pages 2 & 7) is that by excluding data on unemployed individuals who used LSD since their unemployment began, I can better explore whether LSD use prior to their job loss can predict psychological distress following their job loss. I hope my minor edits have helped clarify this in my revision.

Thank you as well for bringing your concerns regarding the payment of participants to my attention. It is important to understand that the NSDUH survey is a long survey with a large array of questions regarding individuals’ drug use, health, and general personal situation. Although I was not involved in the data collection (my manuscript makes secondary use of the data), I assume that the $30 compensation was necessary to incentivize enough respondents to participate. That being said, I do not think compensation for participation reduced the authenticity of the data (and as a result my findings) because, as reported in the article, participants were randomly sampled across the United States. This means that financially-motivated participants with little desire for answering the survey questions honestly could not self-select into the study (as would be the case for studies collecting data via online platforms like MTurk or Prolific). Furthermore, considering that participants would be unaware of my particular interest in testing the link between LSD use and psychological distress following job loss, it is unclear to me how their response patterns would explain the current study’s findings. Although this might be the case for experimental studies using overly salient experimental conditions, my findings make secondary use of a large, openly available dataset. Please let me know if you have additional concerns regarding this point.//

Reviewer #2: The manuscript is technically sound with conclusions clearly demonstrable from the data presented, the statistical analysis is rigorous and appropriate, the data is from a publicly available dataset, and is written in fully intelligible standard English.

//Author Response: Thank you very much.//

---

## [Decision Letter · Decision Letter 1]

22 May 2024

Does LSD confer lasting psychological resilience? An investigation of naturalistic users experiencing job loss

PONE-D-23-22708R1

Dear Dr. Korman,

We’re pleased to inform you that your manuscript has been judged scientifically suitable for publication and will be formally accepted for publication once it meets all outstanding technical requirements.

Kind regards,

Anthony A. Olashore, MD, PhD.

Academic Editor

PLOS ONE

**Comments to the Author**

1. If the authors have adequately addressed your comments raised in a previous round of review and you feel that this manuscript is now acceptable for publication, you may indicate that here to bypass the “Comments to the Author” section, enter your conflict of interest statement in the “Confidential to Editor” section, and submit your "Accept" recommendation.

Reviewer #3: (No Response)

2. Is the manuscript technically sound, and do the data support the conclusions?

Reviewer #3: Yes

3. Has the statistical analysis been performed appropriately and rigorously? 

Reviewer #3: Yes

4. Have the authors made all data underlying the findings in their manuscript fully available?

Reviewer #3: Yes

5. Is the manuscript presented in an intelligible fashion and written in standard English?

Reviewer #3: Yes

6. Review Comments to the Author

Reviewer #3: A possible limitation is the average or unemployment duration of Job loss iin relation to the distress I feel it will add more to the rich manuscript. The limitations were well elucidated

7. PLOS authors have the option to publish the peer review history of their article (what does this mean?). If published, this will include your full peer review and any attached files.

Reviewer #3: No

---

## [Editor Report · Acceptance letter]

27 May 2024

PONE-D-23-22708R1 

PLOS ONE

Dear Dr. Korman, 

I'm pleased to inform you that your manuscript has been deemed suitable for publication in PLOS ONE. Congratulations! Your manuscript is now being handed over to our production team.

Kind regards, 

on behalf of

Dr. Anthony A. Olashore 

Academic Editor

PLOS ONE